# Structural and Functional Differences in the Gut and Lung Microbiota of Pregnant Pomona Leaf-Nosed Bats

**DOI:** 10.3390/microorganisms13081887

**Published:** 2025-08-13

**Authors:** Taif Shah, Qi Liu, Guiyuan Yin, Zahir Shah, Huan Li, Jingyi Wang, Binghui Wang, Xueshan Xia

**Affiliations:** 1State Key Laboratory of Conservation and Bio-Resources in Yunnan, Yunnan University, Kunming 650091, China; taifshah@yahoo.com; 2Yunnan Province Key Laboratory of Public Health and Biosafety, School of Public Health, Kunming Medical University, Kunming 650500, China; 3College of Veterinary Sciences, The University of Agriculture, Peshawar 25130, Pakistan; drzahir@aup.edu.pk

**Keywords:** Pomona leaf-nosed bats, gut, lung, microbiota, functional annotation

## Abstract

Mammals harbor diverse microbial communities across different body sites, which are crucial to physiological functions and host homeostasis. This study aimed to understand the structure and function of gut and lung microbiota of pregnant Pomona leaf-nosed bats using V3-V4 16S rRNA gene sequencing. Of the 350 bats captured using mist nets in Yunnan, nine pregnant Pomona leaf-nosed bats with similar body sizes were chosen. Gut and lung samples were aseptically collected from each bat following cervical dislocation and placed in sterile cryotubes before microbiota investigation. Microbial taxonomic annotation revealed that the phyla Firmicutes and Actinobacteriota were most abundant in the guts of pregnant bats, whereas Proteobacteria and Bacteroidota were abundant in the lungs. Family-level classification revealed that Bacillaceae, Enterobacteriaceae, and Streptococcaceae were more abundant in the guts, whereas Rhizobiaceae and Burkholderiaceae dominated the lungs. Several opportunistic and potentially pathogenic bacterial genera were present at the two body sites. *Bacillus*, *Cronobacter*, and *Corynebacterium* were abundant in the gut, whereas *Bartonella*, *Burkholderia*, and *Mycoplasma* dominated the lungs. Alpha diversity analysis (using Chao1 and Shannon indices) within sample groups examined read depth and species richness, whereas beta diversity using unweighted and weighted UniFrac distance metrics revealed distinct clustering patterns between the two groups. LEfSe analysis revealed significantly enriched bacterial taxa, indicating distinct microbial clusters within the two body sites. The two Random Forest classifiers (MDA and MDG) evaluated the importance of microbial features in the two groups. Comprehensive functional annotation provided insights into the microbiota roles in metabolic activities, human diseases, signal transduction, etc. This study contributes to our understanding of the microbiota structure and functional potential in pregnant wild bats, which may have implications for host physiology, immunity, and the emergence of diseases.

## 1. Introduction

Mammalian microbiota is highly diverse, site-specific, and closely associated with host tissues. Microbial populations at various anatomical sites play crucial roles in physiological processes, including digestion, immune system development, and protection against pathogens [1,2]. The gut microbiota in humans, animals, and bats, which contains trillions of microbes, plays an essential role in digestion, vitamin synthesis, immune regulation, and host defense against pathogens [3,4]. Apart from its functional importance, lung microbiota may influence respiratory health, immune tolerance, and host susceptibility to zoonotic diseases [5]. Therefore, a balance between the host and microbiota is essential because any microbial disturbance along the gut–lung axis during pregnancy can result in systemic disorders [1].

Studies on bat ecology have demonstrated that host physiology and environment have a significant impact on gut microbiota structure [4]. A study reported substantial differences in the gut microbiota of pregnant women, including an increase in the phyla Actinobacteria and Proteobacteria during the third trimester [6]. In another study, pregnant women in the first trimester of pregnancy showed higher levels of the phylum Proteobacteria, whereas those with high glycemic index diets had lower levels of Bacteroidetes and Actinobacteria [7]. Longitudinal research has demonstrated dynamic changes in the gut microbiota during pregnancy, with a reduction in Bacteroidetes and an increase in Firmicutes and Actinobacteria, indicating a shift in microbial populations that favors energy storage and metabolic changes necessary for fetal growth [8]. In addition, significant differences in the gut microbiota of various pregnant animal models have been reported. For example, a study reported that pregnant Tibetan macaques (*Macaca thibetana*) had higher levels of Proteobacteria [9], which might be associated with energy metabolism and enhanced nutrient absorption. A cross-sectional study reported that maternal glucocorticoid load in Assamese macaques (*Macaca assamensis*) during early pregnancy leads to gut microbiota dysbiosis, characterized by reduced bacterial richness and an increase in pro-inflammatory taxa [10]. Microbial dysbiosis in mammal guts has been linked to many pathological states such as asthma, chronic obstructive pulmonary disease, and pneumonia [3]. Microbial secondary metabolites, short-chain fatty acids (including acetate, butyrate, and propionate), cytokines, and immune cells play immunomodulatory roles in local and distant pulmonary responses. Conversely, pulmonary infections can influence gut microbiota through cytokine signaling and changes in gastrointestinal motility [5].

Bats possess exceptional flight abilities that allow them to occupy a wide range of ecological niches. Over 1400 bat species exhibit a range of dietary habits, including carnivory, frugivory, insectivory, nectarivory, and sanguivory, which influence their ecological roles as pollinators and seed dispersers [11]. Flying mammals exhibit exceptional longevity for their body size, characterized by improved immune tolerance to a range of pathogens and disorders [12]. Such characteristics piqued our interest in investigating the microbiota structure and its impact on host pregnancy. Bat pregnancy, which involves energy-demanding processes such as flight and torpor entry, may trigger specific microbial adaptations [13]. A similar study reported that diet is a primary determinant shaping gut microbiota among Neotropical bats, resulting in distinct microbial profiles associated with frugivory, insectivory, and nectarivory [14]. Subsequent studies on insectivorous bats have revealed seasonal and geographical differences in microbiota profiles [15].

The Pomona leaf-nosed bat (*Hipposideros pomona*) is a small insectivorous bat species found in the tropical and subtropical forests of Southeast Asia [16]. Like other members of the genus *Hipposideros*, the bat is characterized by a distinctive leaf-like nose and prefers to roost in caves and hollow trees. These bats play an important role in maintaining ecological balance by acting as natural predators of insects, including agricultural pests, thereby reducing the need for pesticides [17]. Owing to global deforestation, industrialization, and increasing human populations, bat populations and their natural habitats have decreased significantly [16]. Despite its ecological role, little is known about the microbiota structure in Pomona leaf-nosed bats, especially in the guts and lungs during pregnancy, a physiological state characterized by intense metabolic changes. Pomona bats harbor significant populations of Firmicutes and Proteobacteria, which can degrade chitin and digest insect exoskeletons [18].

Mammals use immune tolerance mechanisms involving the gut–lung axis to improve systemic immune responses and promote respiratory well-being [1,5]. Multiple factors influence microbiota composition, including the nest’s environment, social interactions, seasonal dietary fluctuations, and contact with soil and aquatic microorganisms. Changes in the microbiota during pregnancy can profoundly impact the outcome of pregnancy and the well-being of offspring [12,13]. Maternal gut microbes from wild bat populations are essential for assessing zoonotic transmission risk, owing to the close association between bats, humans, and other mammals [13].

The current study aims to fill knowledge gaps by describing the microbiota structure inhabiting the guts and lungs of pregnant Pomona leaf-nosed bats. Microbiota functional annotation has enhanced our understanding of infectious diseases and their implications for public health, providing a foundational framework for zoonotic pathogen surveillance and evaluation of risks associated with spillover.

## 2. Materials and Methods

### 2.1. Site Selection and Bat Collection

More than 350 bats were captured using mist nets placed at the entrances of caves, holes, and tunnels during their active periods (i.e., emergence or return flights) in several locations in Kunming city of Yunnan province, China. Nine pregnant Pomona leaf-nosed bats with similar body sizes were chosen from the population sample for microbiota investigation. Preliminary determination of pregnancy was made using non-invasive palpatory procedures applied over the abdomen, the results of which indicated the presence of a distended abdomen. Postmortem examinations after cervical dislocation confirmed the presence of a developing fetus. Gut and lung tissue samples were aseptically collected from each bat following cervical dislocation, placed in sterile cryotubes, and stored in a −80 °C deep freezer for microbial investigation.

### 2.2. DNA Extraction and Pomona Leaf-Nosed Bat Species Confirmation

Molecular and morphological techniques were used to identify the bat species. The external morphology assessment involved taxonomic features, such as forearm length, ear morphology, nose structure, and fur color. Phenotypic traits were compared with known taxonomic features specific to Southeast Asian bats. The Pomona leaf-nosed bat species was further confirmed via PCR, in which genomic DNA from 16 to 20 mg of lung tissue was isolated and lysed using the QIAamp Mini DNA Kit protocol (Qiagen, Hilden, Germany). PCR amplification was performed using mitochondrial *cytochrome b* (*Cytb*) gene-specific primers (F:5′-CGAAGCTTGATATGAAAAACCATCGTT-3′, R:5′-GGAATTCATCTCTCCGGTTTACAAGA-3′). The reaction was conducted in a 25 µL setup containing 12.5 µL of Master Mix, 0.5 µM of the reverse and forward primers, 2 µL of the DNA template, and nuclease-free water. The thermal cycling parameters were set as follows: initial denaturation for 4 min at 95 °C, followed by 32 cycles of 30 s at 94 °C denaturation, 45 s at 52 °C annealing, and 120 s at 72 °C extension, with a final extension at 72 °C for 10 min. The 1190 bp PCR product was validated on a 1% agarose gel before Sanger bidirectional sequencing. The resulting consensus sequencing reads were BLASTed against the NCBI database (sequence similarity ≥ 98%) for species identification.

### 2.3. Microbial DNA Extraction and V3-V4 16S rRNA Gene Amplification

Microbial DNA was extracted from gut and lung tissue samples according to the protocol of the QIAamp DNA Host-free Microbiome Kit (Qiagen, Hilden, Germany). After determining DNA concentration and purity, the V3-V4 16S rRNA gene hypervariable regions were amplified with the gene-specific primer set 338F (5′-ACTCCTACGGGAGGCAGCA-3′) and 806R (5′-GGACTACHVGGGTWTCTAAT-3′). The PCR reactions were carried out in 25 μL volumes, which included 12.5 μL KAPA ReadyMix (Roche, Indianapolis, IN, USA), 1 μL each of F and R primers, and 2 μL of the DNA template, with the remaining volume completed with nuclease-free water. The PCR cycling conditions included an initial denaturation at 95 °C for 3 m, followed by 30 cycles (denaturation at 95 °C for 30 s, annealing at 55 °C for 30 s, and extension at 72 °C for 30 s), and a final extension at 72 °C for 7 m. The resulting amplicons were analyzed using a 1% agarose gel, purified using AMPure XP beads, and quantified using the Qubit dsDNA HS assay (Thermo Fisher Sci., Waltham, MA, USA).

### 2.4. DNA Library Preparation, Sequencing, and Bioinformatics Analysis

Purified amplicons of the V3-V4 16S rRNA gene were used to construct DNA libraries using the Nextera XT DNA Library Preparation Illumina Kit in conjunction with the Nextera XT Index Kit (Illumina Inc., San Diego, CA, USA). These libraries were then pooled in equimolar concentrations and sequenced on the Illumina MiSeq 2500 platform, generating high-throughput data for subsequent analyses. Raw sequencing data were processed using the QIIME2 pipeline (v. 2) [19]. The QIIME demux plugin was used to evaluate the quality of reads from the original FASTQ files before trimming. The DADA2 plugin [20] was used to perform denoising, dereplication, pairing of paired-end reads, chimera removal, and error correction of sequencing data, thereby generating amplicon sequence variants (ASVs).

All ASVs were aligned using QIIME2 and MAFFT alignment (v. 7.526) tools and then masked to remove hypervariable positions. Phylogenetic diversity analysis was performed using QIIME2 Phylogeny FastTree2 (which generates both rooted and unrooted trees). Each ASV taxonomy was assigned by a Naive Bayes classifier trained on the SILVA 138 reference database [21]. Various plots show the relative microbial abundances at different taxonomic levels. Alpha diversity indices (Chao1, Simpson, Shannon, and observed ASVs) were used to assess microbial richness and evenness, while beta diversity was measured using Euclidean and Bray–Curtis dissimilarity metrics and weighted and unweighted UniFrac distances to evaluate differences in microbial communities between the gut and lung sample groups. The QIIME2 output dataset, which included a feature table, taxonomy annotation, and distance matrices, was exported to the R program (v. 4.5.0) for visualization. The phyloseq R (v. 4.5.0) package organizes and analyzes sample metadata, taxonomy, and phylogenetic trees. Statistical analyses were performed using the vegan R (v. 4.5.0) package, which included principal coordinate analysis (PCoA) and PERMANOVA tests to compare the sample groups. PCoA plots were used to compare the microbial community structures of the gut and lung samples. LEfSe test (using a *p*-value < 0.05 and an LDA score greater than 2) identified differentially abundant microbial taxa to ensure biological relevance. Taxonomic compositions and relative abundances are depicted using bar plots, heat maps, and other diversity plots. PERMANOVA and Kruskal–Wallis tests were used to compare the gut and lung microbiota of the Pomona bats, respectively. The Random Forest classifier, comprising 100 random forests with 500 trees, enabled the accurate classification and identification of key microbial taxa that distinguished the two groups. PICRUSt2 software (v. 2.0) [22] was used to infer the functional features of microbiota. The KEGG Orthologue database served as the reference for predicting gene family abundances, whereas MetaCyc (MetaCyc.org) database facilitated pathway abundances between the two sample groups.

## 3. Results

To explore microbiota composition, this study utilized V3-V4 16S rRNA gene high-throughput sequencing libraries from 18 tissue samples (9 guts and 9 lungs) from nine pregnant wild Pomona leaf-nosed bats. Overall, the two sample group libraries yielded 2,178,028 clean reads (with an average of 1,109,014) and 1,999,481 clean tags (average of 999,742), with a 95.6% average Q-score of 30% (Appendix A). All quality sequences in the guts and lungs of pregnant leaf-nosed bats were attributed to 4035 ASVs (Appendix A). Of these ASVs, 1916 were gut-specific, 1643 were lung-specific, and 476 were shared (Figure 1a). More specifically, preg-gut-5 had the highest ASV count (399), followed by preg-gut-2 (263) and preg-lung-9 (216). The Venn diagram shows the unique and shared ASVs among gut and lung samples (Figure 1b).

According to the relative abundance profiles, ASV-1, ASV-4, ASV-5, ASV-6, ASV-9, and ASV-14 were enriched in the guts, whereas ASV-2, ASV-3, ASV-5, ASV-7, and ASV-11 were more abundant in pregnant Pomona bat lungs (Figure 2a). Notably, ASV-5 was shared by both groups, implying its role in systemic microbial dynamics. Furthermore, microbial taxonomic analysis revealed four dominant phyla that were consistently present in the gut and lungs of pregnant bats. Phyla Firmicutes and Actinobacteriota were the most prevalent in the guts of pregnant bats; meanwhile, Proteobacteria and Bacteroidota were abundant in the lungs of Pomona leaf-nosed bats (Figure 2b). Bacterial classes Bacilli, Gammaproteobacteria, and Actinobacteria were more abundant in the guts; however, Alphaproteobacteria, Rhodothermia, and Bacteroidia were more abundant in the lungs (Appendix A), indicating niche-specific microbial adaptations. Family-level classification revealed Bacillaceae, Enterobacteriaceae, and Streptococcaceae were more abundant in the guts, whereas Rhizobiaceae, Burkholderiaceae, and Balneolaceae dominated the lungs (Figure 2c). These compositional differences highlight the distinct microbial textures at the two body sites during pregnancy. The two body sites had several opportunistic and potentially pathogenic bacterial genera. *Bacillus*, *Cronobacter*, and *Corynebacterium* were abundant in the guts, whereas *Bartonella*, *Burkholderia*, and *Mycoplasma* were more prevalent in the lung samples (Figure 2d), which may indicate a higher risk of dysbiosis or opportunistic infections of pregnant Pomona leaf-nosed bats. Details of the abundant taxonomic profiles in the guts and lungs of Pomona leaf-nosed bats are shown in Appendix A.

### 3.1. Alpha and Beta Diversity Within Gut and Lung Microbiota

To assess microbial diversity (alpha diversity) within sample groups, we examined read depth, species richness, Chao1 index, and Shannon index for gut and lung microbiota. Overall, the number of sequenced reads was consistent across the sample types, ranging from 53,700 to 111,952 reads per sample (Figure 3a), indicating adequate coverage for diversity analyses. The species richness (observed ASVs) and Chao1 estimator, which account for unseen taxa, revealed that gut samples had higher diversity than lung samples. The richness values in the gut samples ranged from 288 to 649, whereas those in the lungs ranged from 217 to 525 (Figure 3b). Similarly, the Chao1 values matched the microbial richness of all samples (Figure 3c), indicating that the sampling depth was adequate. The Shannon diversity index revealed greater variation between sample types, with lung samples ranging from 1.4 to 5.37 and gut samples ranging from 0.87 to 5.65 (Figure 3d). Although sample gut-1 had a low Shannon index (0.871), most guts were more evenly distributed and diverse than lungs. The mean Shannon index for gut samples was 4.34 ± 1.51, compared to 4.03 ± 1.25 for the lungs (Appendix A), indicating greater microbial diversity in the guts.

PCoA was used to compare the beta diversity between the two groups using both unweighted and weighted UniFrac distance metrics. Weighted UniFrac PCoA, which considers phylogenetic relationships and relative abundance, revealed a different clustering pattern. PCoA1 and PCoA2 showed a higher proportion of variance (34.1% and 28.8%) (Figure 4a), indicating abundance-weighted differences between the two groups. Lung samples had a wider spread along PCoA1, with lung-3 and lung-4 scoring strongly positive; however, gut samples clustered with more consistent negative or near-zero values (Appendix A). The unweighted UniFrac PCoA, which considers the presence or absence of taxa, revealed a distinct difference in the community structure between the lung and gut microbiota. The first coordinate (PCoA1) with 21.2% microbial variation showed obvious differences between the two groups (Figure 4b). In contrast, the second coordinate (PCoA2, 11.2%) distinguished intra-group variations, particularly among the lung samples. This pattern suggests microbial community structure in the two body sites varies with the presence or absence of specific taxa.

The heatmaps show that the Bray–Curtis matrix effectively distinguished between the microbiota structures of the two groups (Figure 4c). Gut samples had low dissimilarity indices, with some samples (i.e., gut-5 and gut-6) exhibiting high similarity (dissimilarity index of 0.3143), indicating consistent microbial communities within the guts. Similarly, lung samples showed internal similarity, with the lowest dissimilarity between lung-5 and lung-7 (0.1082). In contrast, comparisons of the gut and lung samples consistently yielded dissimilarity values greater than 0.9 (Appendix A), indicating a significant compositional difference between the two body sites. Furthermore, the Euclidean distance matrix revealed a distinct clustering of gut and lung microbiota profiles (Figure 4d), which supports the separation observed in the Bray–Curtis analysis. Within the guts, samples had low to moderate Euclidean distances, with the smallest distance (0.4372) found between gut-2 and gut-3, indicating closely related microbial composition. Gut-5 and gut-6 also had a high similarity (0.3197), indicating strong internal consistency. Lung samples also showed tight clustering, with the smallest distances observed between lung-1 and lung-8 (0.1154) and lung-5 and lung-7 (0.1180), indicating homogeneity in their microbial texture. In contrast, inter-group distances between gut and lung samples were consistently high, as observed between gut-3 and lung-1 (1.412) or gut-7 and lung-4 (1.395) (Appendix A), indicating significant differences in the microbial communities. These patterns highlight strong site-specific microbial signatures and indicate a minimal overlap between the gut and lung microbiota.

### 3.2. Differentially Abundant Taxonomic Biomarkers Reflecting Microbial Dysbiosis

LEfSe analysis revealed significant enrichment in various taxa (LDA score > 2.5, *p* < 0.05), indicating distinct microbial communities between the two groups (Figure 5a). The most prominent biomarkers observed in the guts were p_Actinobacteria (LDA 4.95, *p* 0.047), c_Actinobacteria (LDA 4.94, *p* 0.047), o_Propionibacteriales (LDA 4.034, *p* 0.02), and f_Nocardioidaceae (LDA 4.02, *p* 0.02). The significantly enriched gut-associated genera were *Leuconostoc* (LDA score 4.004, *p* 0.01), *Aeromicrobium* (LDA 3.84, *p* 0.01), *Turicibacter* (LDA 3.23, *p* 0.03), and *Fusibacter* (LDA 3.27, *p* 0.03). Other notable gut-associated taxa included f_Micrococcaceae, f_Saccharimonadaceae, g_*Virgibacillus*, and the *Nitrospira* lineage (e.g., g_*Nitrospira*, f_Nitrospiraceae, o_Nitrospirales), with LDA scores above 2.6 and *p*-values of 0.001 to 0.030. In contrast, the lung microbiota was enriched in f_Rhizobiaceae (LDA score 5.11, *p* 0.002), c_Alphaproteobacteria (LDA 5.11, *p* 0.002), and g_*Bartonella* (LDA 5.10, *p* 0.002), indicating a strong proteobacterial signature. Additional lung-associated biomarkers included s_*Mycoplasma coccoides* (LDA score 4.64, *p* 0.004), g_*Marinomonas* (LDA 3.57, *p* 0.01), and f_Methylopilaceae (LDA 3.35, *p* 0.01). The genera *Streptococcus*, *Anoxybacillus*, *Comamonas*, *Devosia*, and *Pseudoalteromonas* were also abundant in the lungs (LDA score 2.5 to 3.5) (Appendix A). Overall, the LEfSe findings and the cladogram results confirmed the microbial abundances and phylogenetic relationships of the detected taxa in the two anatomical sites (Appendix A).

### 3.3. RandomForest Feature Importance

The Random Forest classification evaluated microbial feature importance using Mean Decrease Accuracy (MDA) (Figure 5b) and Mean Decrease Gini (MDG) (Figure 5c) in the guts and lungs of pregnant Pomona leaf-nosed bats. Patescibacteria emerged as the most influential taxon, with the highest MDA (9.73) and MDG (1.703) scores for differentiating the microbiota structure in the guts and lungs of Pomona bats. Actinobacteriota (MDA 5.27, MDG 0.87), followed by Proteobacteria (MDA 3.67, MDG 0.59) and Bacteroidetes (MDA 3.33, MDG 0.495), made significant contributions to model accuracy. Other notable microbial contributors included Nitrospirota, Verrucomicrobiota, and Firmicutes, with MDA and MDG values greater than one, indicating substantial differentiation. In contrast, the phyla Synergistota, Bdellovibrionota, and Spirochaetota had moderate MDA scores, and their MDG values were significantly lower, indicating a lesser role in classification. Deinococcota had low MDA (0.36) and high MDG (0.45) values, which may contribute more to node homogeneity than overall model accuracy (Appendix A).

### 3.4. Microbiota Functional Annotation

The PICRUST2-based KEGG pathway annotation classified 435 genes into seven functional groups: metabolic pathways (165 genes), human diseases (87), organismal systems (85), environmental information processing (36), cellular processes (35), and genetic information processing (27) (Figure 6a). Additionally, enriched KEGG pathway-annotated genes provided a comprehensive understanding of their biological functions. The highest representation was observed for the biosynthesis of secondary metabolites and signal transduction (each with 28 genes). The endocrine system, glycan biosynthesis, and metabolism were closely monitored, with 21 genes highlighting various metabolic and regulatory roles. Other notable groups included the immune system (20 genes), lipid metabolism (17 genes), and several disease-related pathways, such as cancers (14 genes) and viral infectious diseases (12 genes) (Figure 6b). We also observed distinct changes in the microbial functions in the gut and lungs of pregnant bats. Bacterial chemotaxis consistently showed the highest abundance (in gut-2 and gut-3), indicating increased microbial motility and environmental responsiveness. This trend was mirrored by an increased expression of signal transduction and membrane transport pathways, implying that the microbiota during pregnancy may be more active in adapting to changes in gut biofilm-related pathways, *E. coli* biofilm formation, *Pseudomonas aeruginosa* biofilm formation, and quorum sensing. Antibiotic resistance pathways, antifolate resistance, and beta-lactam resistance were common throughout pregnancy, with significant increases in gut-2 and gut-3. Carbon metabolism, glycolysis, and oxidative phosphorylation pathways were consistently represented, with slight enrichment in lung-3 and lung-4. Secondary metabolite-related pathways, such as polyketide backbone biosynthesis and non-ribosomal peptide biosynthesis, showed varying trends between the two sample groups (Figure 6c).

## 4. Discussion

Knowledge of the microbiota structure and diversity at different anatomical sites is crucial for elucidating the interactions between the host and microorganisms, as well as the resultant determinants of host health. High-throughput V3-V4 16S rRNA gene sequencing revealed a unique microbiota structure in the guts and lungs of pregnant Pomona leaf-nosed bats. Sequencing of the 18 tissue samples yielded 2,178,028 clean reads with an average Q-score of 30% (95.6%), indicating high-quality sequencing results. Among the 4035 ASVs, 1916 were unique to the guts, 1643 to the lungs, and 476 were shared, indicating unique microbial communities within the same chiropteran host [23,24]. These findings enhance our understanding of tissue-specific microbiota dynamics during pregnancy and may have far-reaching implications for immune responses and vertical transmission of pathogens.

The relative abundance profiles of ASVs in pregnant Pomona bats showed clear microbial differentiation between the gut and lungs, highlighting the presence of niche-specific bacterial populations [25]. Taxonomic classification revealed four dominant bacterial phyla: Firmicutes and Actinobacteriota in the gut, and Proteobacteria and Bacteroidetes in the lungs. The phylum-level segregation mirrors the differential functional demands and immune status of the two body sites [25]. Differences at the class level, manifested by a higher abundance of Bacilli, Gammaproteobacteria, and Actinobacteria in the gut and Alphaproteobacteria, Rhodothermia, and Bacteroidia in the lungs, highlight habitat-specific adaptations that are presumably governed by factors such as oxygen tension and nutrient availability. Similar trends have been reported in studies on other bat species, where differences in the lung and gut microbiota are maintained during physiological stressors such as gestation [25]. Bacillaceae, Enterobacteriaceae, and Streptococcaceae are more abundant in the guts, while Rhizobiaceae, Burkholderiaceae, and Balneolaceae dominate the lungs, implying functional compartmentalization relevant to digestive efficiency and respiratory homeostasis. Moreover, the presence of potentially pathogenic genera, such as *Bacillus*, *Cronobacter*, and *Corynebacterium*, in the guts, and *Bartonella*, *Burkholderia*, and *Mycoplasma* in the lungs, raises concerns regarding the immunological fitness and susceptibility to opportunistic infections. These microbial trends are consistent with observations that pregnancy alters host–microbe interactions, potentially increasing the risk of dysbiosis and disease [26]. These findings highlight the complex and dynamic nature of bat microbiota as well as the urgent need for tissue-specific monitoring to evaluate maternal health and disease risk in wild chiropteran populations.

The alpha diversity indices, including the Chao1 richness, Shannon diversity, and observed ASVs, revealed variations in the gut and lung compartments. The gut samples had consistently higher species richness, with ASV counts ranging from 288 to 649, compared to the lung samples, which showed counts ranging from 217 to 525. This tendency toward higher Chao1 indices, encompassing both observed and rare taxa, corroborated these observations and ensured a sufficient sampling depth for all tissue samples. These observations are consistent with the literature, which states that the gut generally provides a higher abundance of microbes owing to its complex nutrient landscape and lowered oxygen tension, thereby supporting diverse microbial colonization [27]. The Shannon diversity indices, which measure richness and evenness, ranged between 0.87 and 5.65 for gut samples and 1.4 to 5.37 for lung samples. While some gut samples, such as gut-1, showed a lack of richness (Shannon index: 0.871), the mean gut diversity (4.34 ± 1.51) was higher than that of the lungs (4.03 ± 1.25). Studies on bats and other mammals have reported that alpha diversity is generally stronger in the gut than in the respiratory tract, where environmental conditions and immune responses suppress microbial colonization [28]. The noted differences in microbial diversity between the gut and lungs may also be influenced by pregnancy, which affects hormonal and immune regulation mechanisms, potentially altering the host microbial tolerance and affecting the dynamics of niche colonization [29]. Additionally, beta diversity indices revealed substantial differences between gut and lung microbial communities. PCA, employing weighted UniFrac distances that incorporated taxonomic abundance and associated phylogenetic relationships, revealed that gut and lung samples formed distinct clusters, with PCoA1 and PCoA2 explaining 34.1% and 28.8% of the variance, respectively. Lungs exhibited a broader dispersion along PCoA1, as was evident in lung-3 and lung-4, indicating a greater inter-individual difference in the lung microbiota. Conversely, the guts demonstrated a close clustering pattern, indicating that the microbial pattern was consistent across individuals. The first principal coordinate, PCoA1, which explained 21.2% of the variance, effectively discriminated between the two groups, whereas intra-group variability, particularly among the lung samples, was addressed by PCoA-2, which accounted for 11.2%. These findings show that the observed differences between gut and lung microbiota are caused by variations in community abundance and the presence or absence of specific taxa, indicating niche-specific selective pressure. The compositional dynamics of the microbiota, as revealed by the Bray–Curtis dissimilarity heatmaps, were further illustrated. Intra-group pairwise comparisons revealed greater similarity between the gut samples, with the lowest observed dissimilarity between gut-5 and gut-6. Lung samples also demonstrated similar close grouping, with lung-5 and lung-7 demonstrating the lowest similarity. In marked contrast, inter-group pairwise comparisons consistently revealed a high dissimilarity value (>0.9), indicating significant differences between the gut and lung. This finding was corroborated by using the Euclidean distance matrix analysis, which provided distinct body site-based clustering of samples. The low Euclidean distances observed between specific groups, particularly gut-2 and gut-3 and lung-1 and lung-8, highlight the internal homogeneity of microbial profiles within each distinct anatomical site. In comparison, the observed distances between different groups were much higher, with the highest disparity between gut-3 and lung-1, highlighting significant physiological and immunological variations intrinsic to different body sites [30]. The anaerobic environment and complex nutrients within the gut regions support a diverse and interconnected microbiota that is involved in various metabolic processes. In contrast, the lung is an oxygen-rich, immunologically demanding environment where metabolic colonization of microbes is stringently regulated to prevent infection [31], holding important implications for host health. The abundance of microbial communities within the guts and lungs may enhance organ-specific immune modulation, metabolic activities, and protection from pathogenic agents. Similarly, changes in microbiota composition during pregnancy may affect maternal adaptation, energy balance, and susceptibility to pathogens [32]. Identifying these patterns in free-living Pomona bats advances our understanding of host–microbe interactions under physiological stress and the ecology of emerging diseases. Collectively, this work shows that the gut and lung microbiota of pregnant Pomona bats demonstrate divergent alpha and beta diversity patterns. The gut microbiota demonstrated higher diversity, while the lung microbiota was characterized by increased compositional constraints and variability.

The identification of differentially abundant taxonomic biomarkers using LEfSe explains the unique arrangements of microbiota and their potential health and disease implications. Notably, the presence of Actinobacteriota in the guts indicates a shift toward bacterial populations engaged in maintaining mucosal homeostasis and carbohydrate metabolism [33]. Lactic acid bacteria, including *Leuconostoc*, *Lactobacillus*, and *Enterococcus*, are important biomarkers of gut health, often associated with a healthy gut microbiota due to their roles in short-chain fatty acid production and suppression of pathogen growth [34]. Importantly, the presence of *Aeromicrobium* and *Fusibacter*, which are typically found in environmental or extreme anaerobic niches, may suggest opportunistic colonization or dysbiosis during pregnancy. The *Nitrospira* lineage (g_*Nitrospira*, f_Nitrospiraceae, o_Nitrospirales) shows a significant abundance of nitrite-oxidizing potential. Their presence in the gut microbiota may indicate compromised nitrogen metabolism, which has been linked to inflammatory and metabolic diseases. The presence of *Bartonella* and *Mycoplasma coccoides* indicates a potentially pathogenic microbial profile, commonly implicated in respiratory infections. In addition, the presence of *Marinomonas* and Methylopilaceae suggests that the lung microbiota may undergo transient colonization. The presence of *Streptococcus* in the lungs further complicates its role, as it can act as a commensal or a pathogen depending on the host’s immunity. These microbial composition differences point towards the ecological differentiation of the gut and lung niches. While the gut is predominantly inhabited by anaerobic, fermentative bacteria involved in digestion and promoting metabolic symbiosis with the host, in contrast, the lung habitat supports facultative/anaerobes capable of surviving in variable oxygen tensions [34].

Random Forest classifiers assess microbial features’ importance via MDA, and MDG provides a solid foundation for understanding the key taxa that distinguish gut and lung microbiota in pregnant Pomona bats. Our findings show that Patescibacteria is the most influential phylum, with the highest MDA and MDG scores, emphasizing its pivotal role in shaping microbial structure and supporting evidence that Patescibacteria plays critical ecological roles in host-associated microbiota [35]. Their metabolic specialization and symbiotic relationships may drive significant functional differentiation in bat microbiota, mirroring larger patterns seen in mammalian hosts. The prominence of Actinobacteriota confirms its well-documented role in microbial community stability and host health. Similarly, Proteobacteria and Bacteroidota make moderate but significant contributions, consistent with their known roles in nutrient cycling and immune system interaction in mammalian hosts. Notably, Nitrospirota, Verrucomicrobiota, and Firmicutes exhibited MDA and MDG values above one, highlighting their significance in distinguishing microbial community composition in the two body sites. Firmicutes ferment complex carbohydrate structures to secrete SCFAs, acetate, butyrate, and propionate, which are essential for regulating gut integrity and mucosal immunity [36]. The lower MDA but moderate MDG values for Deinococcota indicate a more nuanced role, possibly contributing more to classification node homogeneity and microbial community cohesion than overall discriminatory power. Synergistota, Bdellovibrionota, and Spirochaetota had moderate MDA but low MDG scores, indicating a minor role in microbial classification. This pattern may indicate niche-specific or transient microbial populations that have a minimal influence on host microbial ecosystem differentiation. The diverse contributions of taxa underscore the intricate interplay between microbial phylogeny, function, and the host environment.

The functional predictions obtained from PICRUSt2 clarify the biological importance of microbiota in pregnant Pomona bats. The dominance of genes involved in metabolic processes is consistent with the significant contribution of microbiota to nutrient processing, which is especially essential for satisfying the mother’s metabolic demands during pregnancy [37]. The high abundance of genes involved in human diseases and organismal systems suggests a dual role for the microbiota in health, disease susceptibility, and complex host–microbe interactions. Signal transduction pathways enable communication and adaptation of microbes, which are essential under the changing conditions posed by pregnancy [38]. The discovery of genes involved in genetic information processing indicates active transcriptional and translational processes, further emphasizing the functional diversity of the microbiota. The enrichment of genes involved in secondary metabolite biosynthesis and signal transduction is especially intriguing. Secondary metabolites, such as polyketides and non-ribosomal peptides, play a crucial role in mediating microbial competition and symbiotic interactions. Similarly, the augmentation of signal transduction pathways aligns with the idea that the microbiota actively orchestrates its dynamic functions to maintain host homeostasis. Pathways involved in the endocrine system, glycan biosynthesis, and metabolism indicate the involvement of microbiota in hormonal modulation and glycan-mediated signaling in the host. These processes have been implicated in fetal development and maternal immune tolerance, thereby providing evidence for recent studies demonstrating that microbial metabolites influence pregnancy outcomes [39]. Functional changes in genes involved in bacterial chemotaxis and motility, especially those in the gut, reflect increased microbial responsiveness as well as the potential for spatial reorganization in the host environment during pregnancy. The simultaneous upregulation of biofilm formation pathways (specifically in *E. coli* and *Pseudomonas aeruginosa*) and quorum sensing illustrates a community-level adaptation that promotes stable microbial consortia and resilience. The frequent appearance of antibiotic resistance mechanisms, such as antifolate and beta-lactam resistance, highlights the likely role of the microbiota in protecting bats from opportunistic infections during gestation. These resistance traits may represent an evolutionary balance between host and microbial survival [34]. This reorganization of functions during pregnancy can impact microbial community dynamics and host physiological responses, ultimately determining the pregnancy outcomes through controlling inflammation, nutrient uptake, and immune regulation. Our findings suggest that the maternal microbiota undergoes extensive functional reorganization, including increased motility, bacterial biofilm formation, antimicrobial resistance, and metabolic adaptations. Further research, including metabolomics and immunological characterization, would help to understand how the microbiota influences pregnancy physiology and disease susceptibility.

The current study has some implications and limitations. The identification of distinct taxonomic biomarkers in the guts and lungs of Pomona bats implies that the microbial communities are closely linked to physiological functions or pregnancy-related hormonal changes. These microbial shifts may impact host health, immune responses, and pathogen susceptibility, offering insights into host–microbe interactions in chiropteran species. Understanding these microbial signatures can aid conservation efforts, zoonotic risk assessments, and the role of microbiota in bat pregnancy. Moreover, some limitations must be acknowledged. Firstly, the small sample size can limit the statistical strength and findings. Second, while LEfSe can identify taxa with relative abundances, it cannot determine causal associations or functional importance; a metagenomics investigation will comprehensively examine microbial functions and their interaction with the host organs. Although LEfSe provides an understandable overview of differentially abundant taxa, future research could supplement such findings with the MaAsLin2 analysis [40] to validate data robustness. Finally, relying on the V3-V4 16S rRNA region limits species-level resolution; thus, full gene sequences will provide more information about potential pathogens relevant to public health.

## 5. Conclusions

This study uses V3-V4 16S rRNA gene high-throughput sequencing to investigate the structural and functional differences in the gut and lung microbiota of pregnant wild Pomona bats. Our findings revealed significant differences in the microbiota structure of two body sites, with the gut microbiota being more diverse than the lungs. Taxonomic and functional analyses revealed microbial signatures unique to each body site, indicating ecological specialization within host tissues. Surprisingly, both body sites contained opportunistic and potentially pathogenic taxa, suggesting a delicate balance in host–microbe interactions during pregnancy, which may influence maternal physiology and pregnancy outcomes. The study establishes a baseline for future research into host–microbiota interactions in wild mammals, emphasizing the ecological significance of gestation-related microbial changes.

## Figures and Tables

**Figure 1 microorganisms-13-01887-f001:**
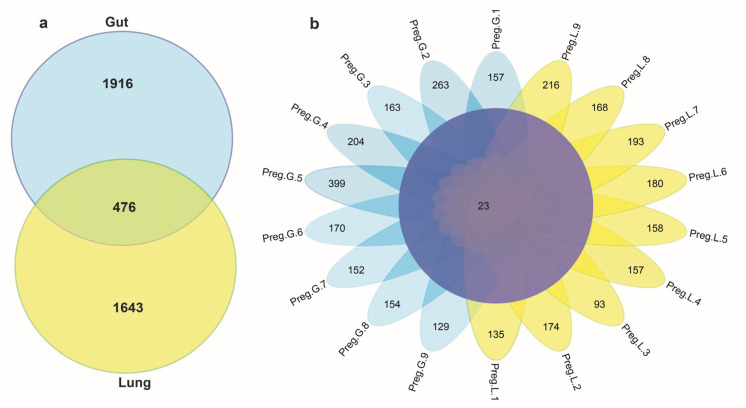
(**a**) The Venn diagram depicts unique and shared ASVs across the two body sites. (**b**) The total number of unique and shared ASVs in each gut and lung sample group.

**Figure 2 microorganisms-13-01887-f002:**
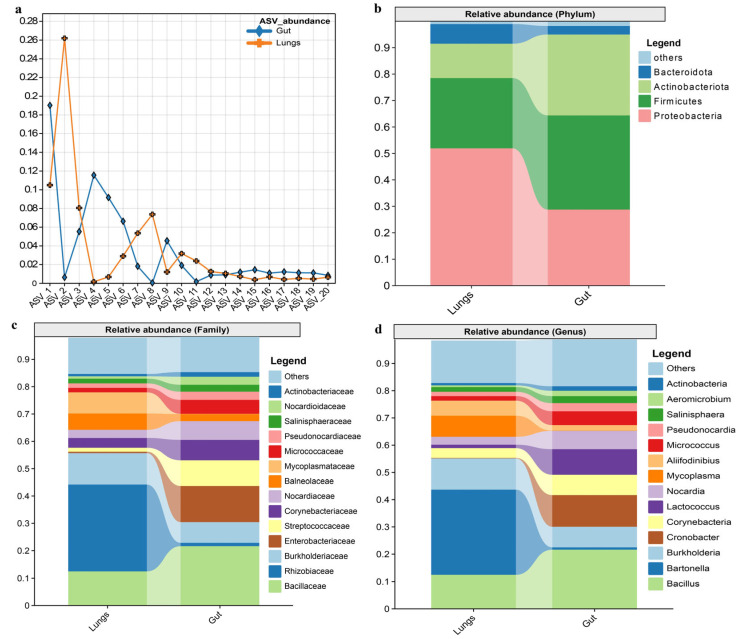
The relative abundance of some key ASVs and taxa in the guts and lungs of pregnant Pomona leaf-nosed bats. (**a**) The abundance of the top twenty ASVs across the two body sites. (**b**) Distribution of phyla, (**c**) families, and (**d**) genera in the guts and lungs of pregnant Pomona bats.

**Figure 3 microorganisms-13-01887-f003:**
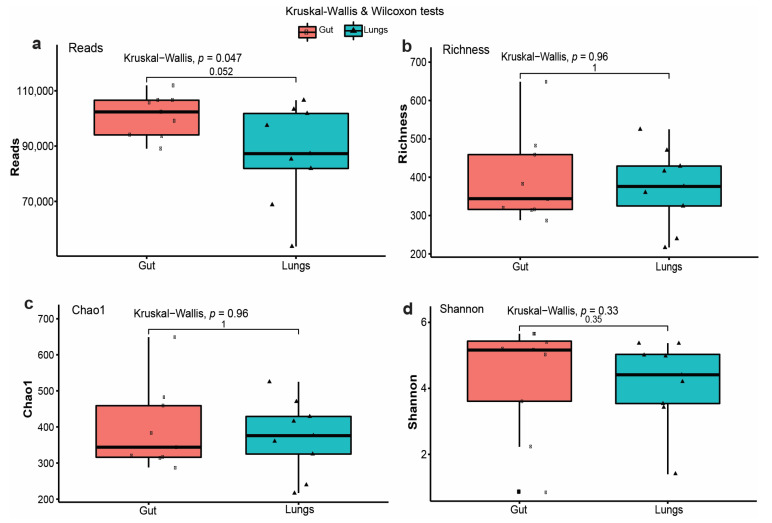
Microbial diversity within groups. (**a**) The number of sequenced reads between the two sample groups. (**b**) Species diversity, (**c**) Chao1 values matched the microbial richness of all samples, and (**d**) the Shannon diversity index revealed greater variability between the two body sites.

**Figure 4 microorganisms-13-01887-f004:**
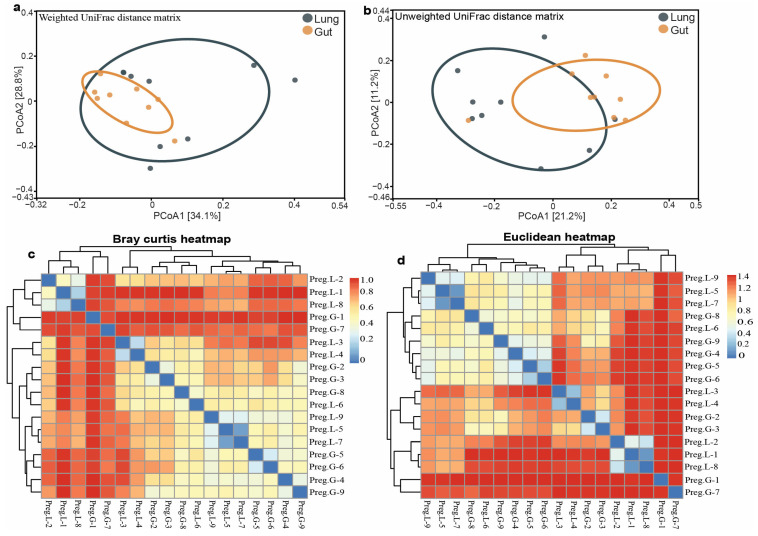
PCoA plots were used to compare beta diversity between the two groups, with both unweighted and weighted UniFrac distance metrics. (**a**) The weighted UniFrac PCoA and (**b**) the unweighted UniFrac PCoA revealed distinct clustering patterns for the two groups. Heatmaps show that the (**c**) Bray–Curtis matrix and (**d**) Euclidean distance matrix effectively separate the microbiota profiles of the two groups.

**Figure 5 microorganisms-13-01887-f005:**
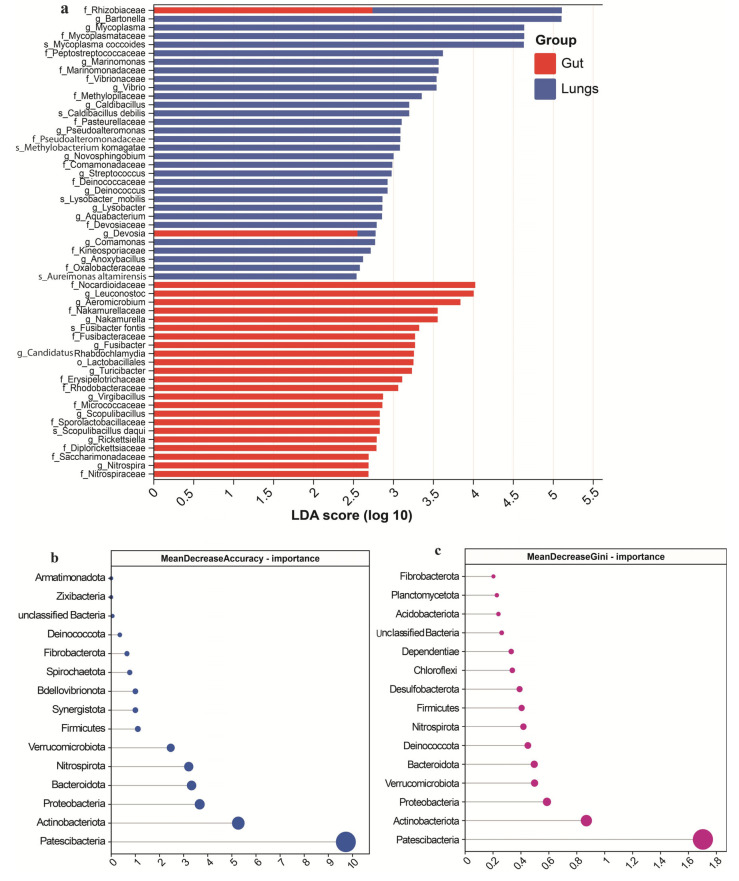
LEfSe identified significantly abundant taxonomic biomarkers, and the Random Forest test revealed feature importance in both groups. (**a**) LEfSe analysis revealed differences in the abundance of taxonomic biomarkers between the two groups. The Random Forest test assessed the importance of microbial features (**b**) MDA and (**c**) MDG in the guts and lungs of pregnant Pomona bats.

**Figure 6 microorganisms-13-01887-f006:**
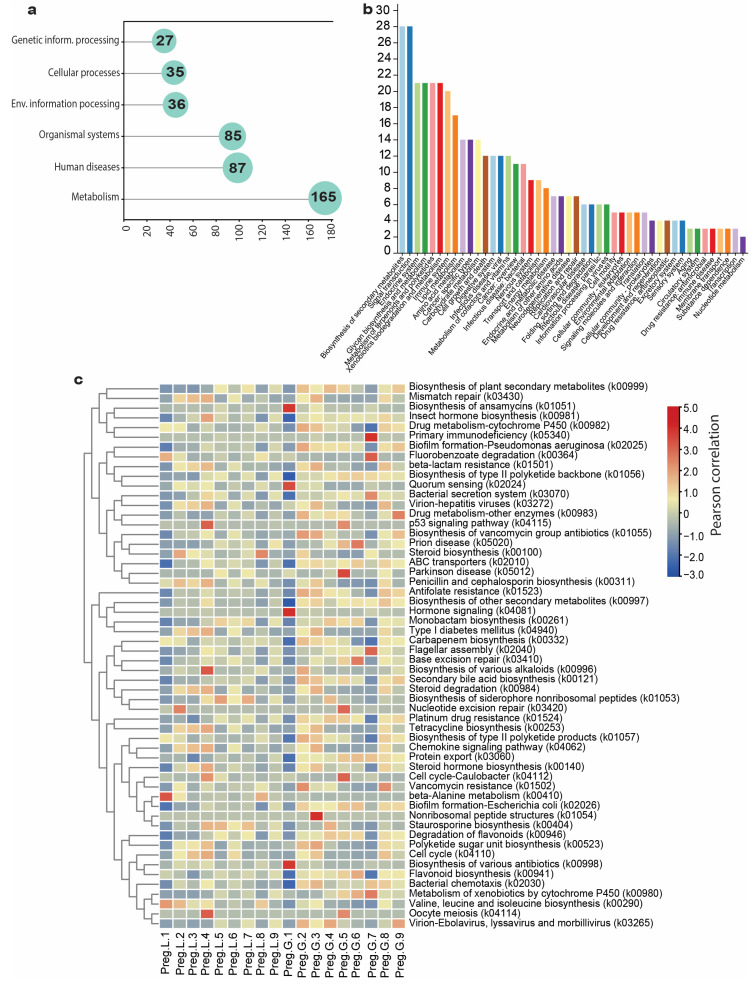
PICRUST2-based functional annotations. (**a**) The PICRUST2 annotation identified seven functional groups. (**b**) The number of genes annotated to various pathways. (**c**) Pearson correlation of the top abundance KEGG pathways in each of the guts and lungs of pregnant Pomona bats.

## Data Availability

The sequencing data for 16S rRNA have been deposited in the Sequence Read Archive (SRA) of the NCBI under BioProject ID: PRJNA1209646 https://www.ncbi.nlm.nih.gov/search/all/?term=PRJNA1209646 (accessed on 8 August 2025).

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
