# Peer review of "Structural and Functional Differences in the Gut and Lung Microbiota of Pregnant Pomona Leaf-Nosed Bats"

_microorganisms, 2025, doi:10.3390/microorganisms13081887_

Round 1

Reviewer 1 Report

Comments and Suggestions for Authors

Title: Structural and functional differences in the gut and lung microbiota of pregnant Pomona leaf-nosed bats

Overall recommendation: Major revision

General Comments

This manuscript presents a relevant and original contribution to the study of microbiome in bat tissues. Overall, it’s an interesting study that contributes valuable insights to the expanding field of microbiome research. I believe the study could be strengthened by incorporating additional references and providing a deeper discussion through comparisons with existing research.

Specific Comments by Section

  1. Abstract:

Lines 15-16: In my opinion there is no need to mention here this kind of details.  

  1. Introduction:
  • Lines 42-45: I suggest that after each statement to add some reference. You are referring to studies related to human microbiota or animal? In particular for bats?
  • Line 49: Again why you did not introduce the studies on bats compared with those on humans?
  • Lines 64-65: Is to quick the passing from the description of the changes during pregnancy to lung microbiota. Rephrase and compare with mammals, in particular with bats.
  • Line 71: Based on which studies? Add some references.
  • Line 87: Add some references.
  • Line 101: Based on which studies? Add some references.

  1. Materials and Methods

Comprehensive but in need of refinement and clarification in several areas.

  • Line 117: I suggest to add a figure with the sampling site/sites.
  • Lines 116-127: How the bats were captured? Where the necropsy was performed? The samples were stored directly to -80C?
  • Lines 164-173: Add references for QIIME2 and DADA2 and to all type of the analysis that was used.

  1. Results:
  • Line 201: Mention V3-V4.

  1. Discussion:

Overall, I recommend focusing the discussion more on comparisons with previous studies and incorporating additional citations.

Overall Recommendation

The study addresses an important topic with an innovative framework. However, the introduction, statistical rigor, and interpretation of findings require improvement to support the conclusions.

Author Response

Dear Editor,

We highly appreciate your efforts and consideration of our manuscript as a possible publication. Please convey our hearty thanks to the handling editor and referees for their constructive and positive comments and suggestions. We have revised this manuscript entitled "Structural and functional differences in the gut and lung microbiota of pregnant Pomona leaf-nosed bats" carefully according to their suggestions. However, if something is missing, we will gladly revise the manuscript. The details of the comments and their answers are given below

Reviewer 1#

General comments

This manuscript presents a relevant and original contribution to the study of microbiome in bat tissues. Overall, it’s an interesting study that contributes valuable insights to the expanding field of microbiome research. I believe the study could be strengthened by incorporating additional references and providing a deeper discussion through comparisons with existing research.

Specific comments by section

Abstract:

Lines 15-16: In my opinion there is no need to mention here this kind of details.

Response: Thanks to the reviewer opinion. Based on his opinion, we have revised and added new texts in the revised version.

Introduction:

Lines 42-45: I suggest that after each statement to add some reference. You are referring to studies related to human microbiota or animal? In particular for bats?

Response: Thanks to the reviewer suggestion, we have modified this section “The gut microbiota in humans, animals, and bats, which contains trillions of microbes, plays an essential role in digestion, vitamin synthesis, immune regulation, and host defense against pathogens [3,4]. Apart from their functional importance, lung microbiota may influence respiratory health, immune tolerance, and host susceptibility to zoonotic diseases [5]. Therefore, a balance between the host-microbiota is essential because any microbial disturbance along the gut-lung axis during pregnancy can result in systemic disorders [1]”.

Line 49: Again why you did not introduce the studies on bats compared with those on humans?

Response: As we know that, both human and bats are both mammals, and share niches at some points in their lives. We find little literatures about guts and lungs microbiota of bats, therefore, we tackle some introduction texts with similar mammals, such as human.

Lines 64-65: Is to quick the passing from the description of the changes during pregnancy to lung microbiota. Rephrase and compare with mammals, in particular with bats.

Response: Thanks, we have revised this section based on the reviewer comments.

Line 71: Based on which studies? Add some references.

Response: Thank for improving our manuscript, we have revised and added new information with updated references, i.e., “Microbial secondary metabolites, short-chain fatty acids (including acetate, butyrate, and propionate), cytokines, and immune cells play immunomodulatory roles in local and distant pulmonary responses. Conversely, pulmonary infections can influence gut microbiota through cytokine signaling and changes in gastrointestinal motility [5] …”

Line 87: Add some references.

Response: Thanks, we have added reference “These bats play an important role in maintaining ecological balance by acting as natural predators of insects, including agricultural pests, thereby reducing the need for pesticides [17]”.

Line 101: Based on which studies? Add some references.

Response: we have thoroughly revised this section, and added new informations, i.e., Mammals use immune tolerance mechanisms involving the gut-lung axis to improve systemic immune responses and promote respiratory well-being [1,5]. Multiple factors influence microbiota composition, including the nest's environment, social interactions, seasonal dietary fluctuations, and contact with soil and aquatic microorganisms. Changes in the microbiota during pregnancy can profoundly impact the outcome of pregnancy and the well-being of offspring [12,13]. Maternal gut microbes from wild bat populations are essential for assessing zoonotic transmission risk, owing to the close association between bats, humans, and other mammals [13].”

Materials and Methods

Comprehensive but in need of refinement and clarification in several areas.

Response: we have thoroughly revised MM section, and added/edited informations.

Line 117: I suggest to add a figure with the sampling site/sites.

Response: we very much respect the reviewer suggestion; however, we have already added sox figures with several sub-figures. The journal requirement, won’t allow us to add additional figures.

Lines 116-127: How the bats were captured? Where the necropsy was performed? The samples were stored directly to -80C?

Response: We thank the reviewer for such valuable comments. We have added “More than 350 bats were captured using mist nets placed at the entrances of caves, holes, and tunnels during their active periods (i.e., emergence or return flights) in several locations in Kunming city of Yunnan province, China. In addition, Postmortem examinations after cervical dislocation confirmed the presence of a devel-oping fetus. Gut and lung tissue samples were aseptically collected from each bat following cervical dislocation, placed in sterile cryotubes, and stored in a -80°C deep freezer for microbial investigation.”

Lines 164-173: Add references for QIIME2 and DADA2 and to all type of the analysis that was used.

Response: We thank you reviewer for raising such crucial points. We have added “QIIME2 pipeline [19]. The QIIME demux plugin was used to evaluate the quality of reads from the original FASTQ files before trimming. The DADA2 plugin [20] …”

Results:

Line 201: Mention V3-V4.

Response: Thank you, we have added “… V3-V4 16S rRNA gene…”, in the revised version.

Discussion:

Overall, I recommend focusing the discussion more on comparisons with previous studies and incorporating additional citations.

Response: we have carefully revised the discussion section after tackling both the reviewers’ comments/suggestions. However, at some points, we failed to add data related to bats, due to limited literatures available in the databases. As we know that, both human and bats are both mammals, and share niches at some points in their lives. We find little literatures about guts and lungs microbiota of bats, therefore, we tackle some discussion texts with similar mammals, such as human, and other animals.

Overall Recommendation

The study addresses an important topic with an innovative framework. However, the introduction, statistical rigor, and interpretation of findings require improvement to support the conclusions.

Response: we very much appreciate the reviewer #1 comments/suggestions. His/her critical reading improved our manuscript. We believe that the article has improved and reached to the journal standard for publication. However, if something is missing, we will gladly revise the manuscript.

Reviewer 2 Report

Comments and Suggestions for Authors

Structural and Functional Differences in the Gut and Lung Microbiota of Pregnant Pomona Leaf-Nosed Bats

Abstract: revise. What is the objective of experimenting?  Provide some information about sampling. Be consistent with writing the bacterial names in italics. Provide some of the significant results with a P value.

Improve the introduction and remove unrelated information. Provide more information regarding bats. What do you expect the difference between pregnant and non-pregnant pats? 

L105-113: Focus more on the objectives; no need to provide results. 

Provide the purpose of determining microbial populations inhabiting the guts and lungs of pregnant Pomona leaf-nosed bats.

L122: Provide more information about the sample collection.

Materials and methods: good description.

Results and figures: good presentation.

Conclusion: revise, add the implications of the study, and the limitations.  

Author Response

Dear Editor,

We highly appreciate your efforts and consideration of our manuscript as a possible publication. Please convey our hearty thanks to the handling editor and referees for their constructive and positive comments and suggestions. We have revised this manuscript entitled "Structural and functional differences in the gut and lung microbiota of pregnant Pomona leaf-nosed bats" carefully according to their suggestions. However, if something is missing, we will gladly revise the manuscript. The details of the comments and their answers are given below

Reviewer 2#

Comments and suggestions for authors: Structural and Functional Differences in the Gut and Lung Microbiota of Pregnant Pomona Leaf-Nosed Bats

Abstract: revise. What is the objective of experimenting? Provide some information about sampling. Be consistent with writing the bacterial names in italics. Provide some of the significant results with a P value.

Response: Of the 350 bats captured using mist nets in Yunnan, China, nine pregnant Pomona leaf-nosed bats with similar body sizes were chosen. Gut and lung tissue samples were aseptically collected from each bat following cervical dislocation and placed in sterile cryotubes before being stored at -80°C for microbiota investigation. In addition, all the genera and species names are written with italic. Moreover, P values are shown on the figures in the result sections.

Improve the introduction and remove unrelated information. Provide more information regarding bats. What do you expect the difference between pregnant and non-pregnant pats?

Response: as we know that both bats and humans are mammals, and share same environments. According to our information, limited literature is available about bats guts and lungs microbiota, therefore, we tackle some introduction section with human perspective because of mammals and their close interaction with each other.

L105-113: Focus more on the objectives; no need to provide results.

Response: we have revised and added “The current study aims to fill knowledge gaps by describing the microbiota structure inhabiting the guts and lungs of pregnant Pomona leaf-nosed bats. Microbiota functional annotation has enhanced our understanding of infectious diseases and their implications for public health, providing a foundational framework for zoonotic pathogen surveil-lance and evaluation of risks associated with spillover”.

Provide the purpose of determining microbial populations inhabiting the guts and lungs of pregnant Pomona leaf-nosed bats.

Response: Thanks to the reviewer comments, we have revised and added such informations in conclusion section, “Our findings revealed significant differences in the microbiota structure of two body sites, with the gut microbiota being more diverse than the lungs. Taxonomic and functional analyses revealed microbial signatures unique to each body site, indicating ecological specialization within host tissues. Surprisingly, both sites contained opportunistic and potentially pathogenic taxa, indicating a delicate balance in host-microbe interactions during pregnancy, which may influence maternal physiology and pregnancy outcomes”.

L122: Provide more information about the sample collection.

Response: we have revised and added “More than 350 bats were captured using mist nets placed at the entrances of caves, holes, and tunnels during their active periods (i.e., during emergence or return flights) in several locations in Kunming, Yunnan, China. Nine pregnant Pomona leaf-nosed bats with similar body sizes were chosen from the population sample for microbiota investigation. Preliminary determination of pregnancy was made using non-invasive palpatory procedures applied over the abdomen, the results of which indicated the presence of a distended abdomen. Postmortem examinations after cervical dislocation confirmed the presence of a developing fetus. Gut and lung samples were aseptically collected from each bat following cervical dislocation, placed in sterile cryotubes, and stored in a -80°C deep freezer for microbial investigation”, in the revised manuscript.

Materials and methods: good description.

Response: We thanks to the reviewer for appreciation.

Results and figures: good presentation.

Response: We thanks to the reviewer for appreciation.

Conclusion: revise, add the implications of the study, and the limitations.

Response: we thank for the reviewer suggestions, we have revised and added study implications and limitations “The current study has some implications and limitations. The identification of distinct taxonomic biomarkers in the guts and lungs of Pomona leaf-nosed bats implies that the microbial communities is closely linked to physiological or pregnancy-related hormonal changes. These microbial shifts may have an impact on host health, immune responses, and pathogen susceptibility, offering insights into host-microbe interactions in chiropteran species. Understanding these microbial signatures can aid conservation efforts, zoonotic risk assessments, and the role of microbiota in bat physiology, particularly pregnancy. Moreover, some limitations must be acknowledged. Firstly, the small sample size can limit the statistical strength and findings. Second, while LEfSe can identify taxa with relative abundances, it cannot identify causal associations or functional importance; a metagenomics investigation will comprehensively examine microbial functions and their interaction with the host organs. Although LEfSe provides an understandable overview of differentially abundant taxa, future research could supplement such findings with MaAsLin2 to validate data robustness. Finally, relying on the V3-V4 16S rRNA region limits species-level resolution; thus, full gene sequences will provide more information about potential pathogens relevant to public health”.

Overall. We very much appreciate the reviewer #2 comments/suggestions. His/her critical reading improved our manuscript. We believe that the article has improved and reached to the journal standard for publication. However, if something is missing, we will gladly revise the manuscript.